# Reinforcement learning with non-ergodic reward increments: robustness via ergodicity transformations

**Dominik Baumann**[*]                                                    *dominik.baumann@aalto.fi*
*Cyber-physical Systems Group*
*Aalto University*
*Espoo, Finland*

**Erfaun Noorani**                                                        *enoorani@umd.edu*
*Department of Electrical and Computer Engineering*
*University of Maryland*
*College Park, MD, USA*

**James Price**                                                           *james.price.2@warwick.ac.uk*
*Department of Mathematics*
*University of Warwick*
*Warwick, United Kingdom*

**Ole Peters**[†]                                                         *o.peters@lml.org.uk*
*London Mathematical Laboratory*
*London, United Kingdom*

**Colm Connaughton**[‡]                                                   *c.connaughton@lml.org.uk*
*London Mathematical Laboratory*
*London, United Kingdom*

**Thomas B. Schön**                                                       *thomas.schon@it.uu.se*
*Department of Information Technology*
*Uppsala University*
*Uppsala, Sweden*

**Reviewed on OpenReview:** *https://openreview.net/forum?id=eakh1Edffd*

## Abstract

Envisioned application areas for reinforcement learning (RL) include autonomous driving, precision agriculture, and finance, which all require RL agents to make decisions in the real world. A significant challenge hindering the adoption of RL methods in these domains is the non-robustness of conventional algorithms. In particular, the focus of RL is typically on the expected value of the return. The expected value is the average over the statistical ensemble of infinitely many trajectories, which can be uninformative about the performance of the average individual. For instance, when we have a heavy-tailed return distribution, the ensemble average can be dominated by rare extreme events. Consequently, optimizing the expected value can lead to policies that yield exceptionally high returns with a probability that

---

[*]Also with the Department of Information Technology, Uppsala University, Uppsala, Sweden.
[†]Also with the Santa Fe Institute, Santa Fe, NM, USA.
[‡]Also with the Department of Mathematics, University of Warwick, Warwick, United Kingdom.

approaches zero but almost surely result in catastrophic outcomes in single long trajectories. In this paper, we develop an algorithm that lets RL agents optimize the long-term performance of individual trajectories. The algorithm enables the agents to learn robust policies, which we show in an instructive example with a heavy-tailed return distribution and standard RL benchmarks. The key element of the algorithm is a transformation that we learn from data. This transformation turns the time series of collected returns into one for whose increments expected value and the average over a long trajectory coincide. Optimizing these increments results in robust policies.

# 1 Introduction

Reinforcement learning (RL) has experienced remarkable progress in recent years, particularly within virtual environments (Mnih et al., 2015; Silver et al., 2017; Duan et al., 2016; Vinyals et al., 2019). However, the seamless transition of RL methods to real-world, e.g., robotics, applications lags behind, primarily due to the non-robust nature of conventional RL approaches (Amodei et al., 2016; Leike et al., 2017; Russell et al., 2015). In addressing this issue, researchers have explored a spectrum of methods from risk-sensitive RL (Prashanth et al., 2022) to robust (worst-case) RL (Pinto et al., 2017). In this paper, we take a step back and look at the optimization objective in RL and how it may, by design, result in non-robust policies. Traditional RL literature, including influential references and introductory textbooks such as the ones by Sutton & Barto (2018); Bertsekas (2019); Powell (2021), typically frames the RL problem as maximizing the expected return, i.e., the expected value of the sum of rewards collected throughout a trajectory. Intuitively, at each time step, an agent shall choose an action that maximizes the return it can expect when choosing this action and following the optimal policy from then onward. While this indeed seems intuitive, it may not result in the desired behavior of individual agents. The expected value is the average over infinitely many rollouts of a policy. If we consider an environment with a heavy-tailed return distribution, the highest expected return may be achieved by an extremely risky policy that receives very high returns in a few cases but fails in all others. Suppose an autonomous car learns a driving policy through RL. At deployment time, when we have a passenger in the car, it does not matter to the passenger whether the policy of the autonomous car receives a high return when averaging over multiple trajectories—a high ensemble-average return could also result from half of the journeys reaching the destination very fast and half crashing and never reaching it. The return in a single instance of a long journey would be negligible if a crash occurred somewhere along the way—and this is the return that would matter to the individual. Thus, the *time average* would be the better choice for an optimization objective in such scenarios.

Nevertheless, existing RL algorithms have demonstrated remarkable performance by optimizing expected returns. Optimizing the time average might require developing entirely new RL algorithms. An alternative is to find a suitable *transformation*. In particular, we seek to find a transformation such that for the increments of transformed returns, the expected value and time average coincide. We will discuss that the increments of such transformed returns will then follow a Brownian motion, i.e., the return grows linearly. Then, we can apply existing RL algorithms and optimize the long-term behavior of individual agents.

The field of risk-sensitive RL (Prashanth et al., 2022) follows a similar approach. In most of risk-sensitive RL, e.g., algorithms using an entropic risk measure, the agents try to optimize the expected value of transformed returns. By learning with transformed returns, the agents can achieve higher performance with lower variance. However, those approaches typically work with fixed transformations. Inspired by Peters & Adamou (2018), we analyze for which dynamics a popular transformation from risk-sensitive RL optimizes the long-term return. Further, we propose an algorithm for learning a suitable transformation when the reward function is unknown, which is the typical setting in RL.

**Contributions.** In this paper, we make the following contributions:

- We relate the shortcomings of the expected return as an optimization criterion to the ergodicity of rewards and illustrate and assess the impact of non-ergodic rewards on RL policies through an intuitive example. This showcases the implications of optimizing for the expected value in such

settings—which we commonly encounter in RL problems—and makes a case for the need for an "ergodicity transformation." Note that this notion of ergodicity is not identical to the notion of ergodic Markov decision processes (MDPs) typically studied in RL, and we formally define our notion of ergodicity in section 2.1 and relate it more explicitly to the notion of ergodic MDPs in section 6.1.

- We propose a transformation that can convert a trajectory of returns into a trajectory for whose increments time average and expected value coincide. This enables off-the-shelf RL algorithms to optimize their long-term return instead of the conventional expected value, resulting in more robust policies without developing novel RL algorithms.

- We demonstrate the performance of this transformation in an intuitive example and—as a proof-of-concept—on standard RL benchmarks. In particular, we show that our transformation indeed yields more robust policies.

## 2 Problem setting

We consider a standard RL setting in which an agent with states $s \in \mathcal{S} \subseteq \mathbb{R}^n$ in the state space $\mathcal{S}$ and actions $a \in \mathcal{A} \subseteq \mathbb{R}^m$ in the action space $\mathcal{A}$ shall learn a policy $\pi : \mathcal{S} \to \mathcal{A}$. Its performance is measured by an unknown reward function $r : \mathcal{S} \times \mathcal{A} \to \mathbb{R}$. The agent's goal is to maximize the accumulated rewards $r(t_k)$ it receives during a trajectory, i.e., the *return $R(T)$* at $t_k = T$,

$$R(T) = \sum_{\tau_k=0}^{T} r(\tau_k), \tag{1}$$

where $r(t_k) \coloneqq r(s(t_k), a(t_k))$. For this, the agent interacts with its environment by selecting actions, receiving rewards, and utilizing this feedback to learn an optimal policy. The RL problem is inherently stochastic, as it involves learning from finite samples, often within stochastic environments and with potentially stochastic policies. In standard RL, we, therefore, typically aim at maximizing the expected value of equation 1 (cf. the "reward hypothesis" stated by Sutton & Barto (2018, p. 53))

$$\mathbb{E}_\pi \left[ \sum_{\tau_k=0}^{T} r(\tau_k) \right]. \tag{2}$$

Nonetheless, this conventional approach may encounter challenges when the return distribution is heavy-tailed. To illustrate this point, we consider an instructive example introduced by Peters (2019).

### 2.1 Illustrative example

Imagine an agent starting with an initial reward of $r(t_0) = 100$ is offered the following game. We toss a (fair) coin. If it comes up heads, the agent wins $50\%$ of its current return. If it comes up tails, the agent loses $40\%$. Mathematically, this translates to

$$r(t_k) = \begin{cases} 0.5R(t_{k-1}) & \text{if } \eta = 1, \\ -0.4R(t_{k-1}) & \text{otherwise,} \end{cases}$$

where $\eta$ is a Bernoulli random variable with equal probability for both outcomes.

When analyzing the game dynamics, we find that the agent receives an expected reward $r(t_k)$ equal to $5\%$ of its current return. Consequently, the expected return for any trajectory length $T$ appears favorable, growing exponentially with $T$:

$$\mathbb{E}[R(T)] = 100 \cdot 1.05^T. \tag{3}$$

However, when we simulate the game for ten agents and 1000 time steps, we find that all of them end up having a return of almost zero (see figure 1a). The reason is that the coin toss game is *non-ergodic*. If the dynamics of a stochastic process are non-ergodic, the average over infinitely many samples may be arbitrarily

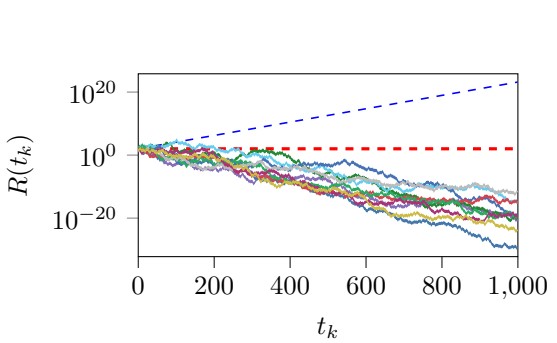
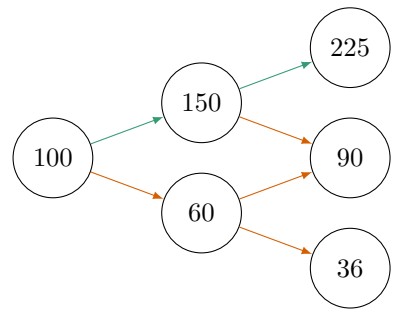

$$100 \xrightarrow{\cdot 1.05} 105 \xrightarrow{\cdot 1.05} 110.25$$

(a) Simulation of the coin toss experiment. *We simulate the game for 1000 time steps and 10 agents. The dashed red horizontal line marks the initial reward of 100, and the dashed blue ascending line the expected value. After 1000 time steps, all agents end up with a lower return than they started with (note the logarithmic scaling of the $y$-axis).*

(b) Possible paths for three iterations of the coin-toss example. *While the average across paths keeps growing, the number of paths that lead to a successful outcome shrinks.*

Figure 1: Simulation and sample paths of the coin-toss example.

different from the average over a single but infinitely long trajectory. We illustrate this phenomenon in figure 1b for three iterations of the game. If we look at the average across iterations, we see that it steadily keeps increasing as predicted by equation 3. However, only one of all possible paths ends up with a return higher than the initial one. In the limiting case, if we simulate infinitely many trajectories of the game, each of finite duration $T$, we obtain a small set of agents that end up exponentially "rich" so that averaging over all of them, i.e., taking the expected value, yields $100 \cdot 1.05^T$. However, if we increase the duration, $T \to \infty$, the set of agents ending up exponentially rich shrinks exponentially to measure zero. That is, if we only simulate one agent for $T \to \infty$ and average over time, we receive a *time average* $\lim_{T\to\infty} \frac{1}{T} \sum_{\tau_k=0}^{T} r(\tau_k) = 0$ almost surely. Hulme et al. (2023) provide a more detailed analysis of the statistical properties of the coin-toss game in their appendix. In appendix A.1, we further show for a fewer number of iterations and more independent runs that there are indeed sample paths that end up with a higher than the initial return.

In RL, ergodicity is typically discussed with respect to the underlying MDP, i.e., we ask whether or not the MDP is ergodic. In this paper, we are concerned with what happens if rewards are non-ergodic. We discuss the relation between those two notions of ergodicity in more detail in section 6.1. To define the notion that we are concerned with properly and connect it explicitly to RL, let us abstract from the coin-toss example and consider an arbitrary discrete-time stochastic process $X$. We can now generate multiple realizations of this process, in the example, by playing the game multiple times. Let $X^{(j)}(t_k)$ denote the value of realization $j$ at time step $t_k$. The process $X$ is *ergodic* if, for any time step $t_k$ and realization $i$,

$$\lim_{N\to\infty} \frac{1}{N} \sum_{j=1}^{N} X^{(j)}(t_k) = \lim_{T\to\infty} \frac{1}{T} \sum_{\tau_k=1}^{T} X^{(i)}(\tau_k) \tag{4}$$

almost surely. The left hand side is $\mathbb{E}[X(t_k)]$, the expected value of $X$ at time $t_k$. The right-hand side is the time average of realization $i$. For an ergodic process, these averages are equal. In the RL setting, we are interested in whether or not the rewards $r(t_k)$ are ergodic, i.e., whether or not

$$\mathbb{E}[r(t_k)] = \lim_{T\to\infty} \frac{1}{T} \sum_{\tau_k=1}^{T} r(\tau_k) = \lim_{T\to\infty} \frac{R(T)}{T} \tag{5}$$

almost surely. For ergodic rewards, maximizing the expected value at each step corresponds to maximizing the long-term growth rate of the return for any given realization. However, as the coin-toss example demonstrates,

when rewards are non-ergodic, optimizing the expected value may yield policies with negative long-term growth rate.

## 2.2 Solving the ergodicity problem

Redefining the optimization objective of RL algorithms may require a complete redesign. Alternatively, we can take existing algorithms and modify the returns to make their increments ergodic. For equation 5 to hold, we need the additive increments of the return to be stationary and independent. As discussed by Peters & Adamou (2018), such processes are Lévy processes. The only Lévy process with continuous sample paths is a Brownian motion with drift (Breiman, 1968, Ch. 12). Thus, following Peters & Adamou (2018) we seek to find a transformation $h(R)$ whose increments $\Delta h$ follow a Brownian motion with drift. In our discrete-time setting, this translates to

$$\Delta h(R(t_{k+1})) = h(R(t_{k+1})) - h(R(t_k)) = \mu + \sigma v(t_k), \tag{6}$$

with drift $\mu$, volatility $\sigma$, and $v(t_k)$ a random variable with finite variance.

In the following, we assess the performance of standard RL algorithms in the coin toss game, with and without a transformation $h$. We then propose an algorithm for learning a transformation $h$ with ergodic increments and relate our findings to risk-sensitive RL.

## 3   RL with non-ergodic dynamics

For the coin toss example, we have already seen empirically that the dynamics are non-ergodic. Optimizing the expected value then yields a "policy" in which the agent decides to play the game, leading to ruin in the long run almost surely. While standard RL algorithms aim to optimize the expected value, they need to approximate it from finitely many samples. Thus, in this section, we evaluate whether a standard RL algorithm indeed proposes a detrimental policy and discuss how we can transform the returns to prevent this. In the version presented in the previous section, the coin toss game offers the agent a binary decision: either play or not. Here, we make the game slightly more challenging by letting the agent decide how much of its current return ("wealth") it invests at each time step. Thus, we have a continuous variable $F \in [0,1]$ and the reward dynamics are

$$r(t_k) = \begin{cases} 0.5FR(t_{k-1}) & \text{if } \eta = 1, \\ -0.4FR(t_{k-1}) & \text{otherwise.} \end{cases} \tag{7}$$

We use the popular proximal policy optimization (PPO) algorithm (Schulman et al., 2017), leveraging the implementation provided by Raffin et al. (2021) without changing any hyperparameters to learn a policy. During training, PPO receives the reward $r(t_k)$ in every iteration. Having trained a policy for $1 \times 10^5$ episodes, we execute it 100 times for 1000 time steps and show the first ten trajectories in figure 2a. We see that all ten agents end up with a return lower than the initial reward of 100. While this could still be caused by a bad choice of agents, it is confirmed by computing statistics over all 100 trajectories. When computing the median of the return after 1000 time steps, we obtain $2.5 \times 10^{-4}$, i.e., the average agent ends up with a return close to zero. The mean over all agent yields 115. That is, a small subset of agents obtains a high return. This confirms the discussion from the previous section. Even if it only approximates the expected value, PPO does learn a policy that leads to ruin for most agents.

One possibility for coping with non-ergodic dynamics is finding a suitable transformation. For the coin toss game, where the dynamics are relatively straightforward and the outcomes are fully known, we can analytically identify an appropriate transformation: the logarithm. While Hulme et al. (2023) provide the technical explanation for why the logarithm is an appropriate transform for the coin-toss game, we here give an intuitive explanation. The dynamics of the coin-toss game are exponential, as can, for instance, be seen from equation 3. Through the logarithm, we basically "linearize" the return dynamics, thus achieving dynamics of the form of equation 6. We subsequently train the PPO algorithm once more with the logarithmic transformation. Specifically, we redefine the rewards as $\tilde{r}(t_k) := \log(R(t_k)) - \log(R(t_{k-1}))$. As before, we run

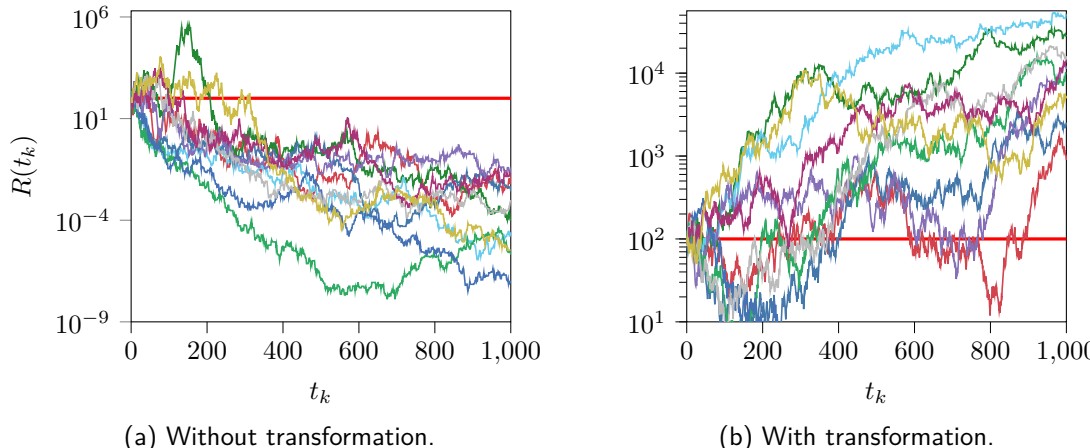

(a) Without transformation.

(b) With transformation.

Figure 2: Learning bet strategies for the adapted coin toss game. *Without transformation, most agents end up losing, while they end up winning with transformation.*

100 experiments for 1000 time steps each and show the first ten trajectories in figure 2b. We see that all agents end up with a significantly higher return than the initial reward. A statistical analysis confirms this observation, yielding a median return of 5645 and a mean of 15 883. Both values substantially surpass those obtained by the agents trained with untransformed returns.

This evaluation underscores that standard RL algorithms may inadvertently learn policies leading to unfavorable outcomes for most agents when dealing with non-ergodic dynamics. Furthermore, it demonstrates that an appropriate transformation can mitigate this. Besides, we see that even the average return is lower for the standard PPO agent, even though this should be what the agent maximizes. The reason for this is that the probability of ending up with a very high reward with a risky policy is non-zero for finite episode lengths but very small. Thus, during training, in some of the $1 \times 10^5$ episodes, the agent will experience some of these very high returns, leading to it assigning high values to those risky policies. When testing on 100 policies, the probability of encountering such a high return rollout is again very low. We can confirm this by evaluating the policy learned with untransformed returns $1 \times 10^6$ times and computing the statistics. Then, we receive a mean return of approximately 48 010, i.e., higher than the mean for the policy learned with transformed rewards.

**Remark 1.** *The quantitative results clearly differ between runs, as the environment and training process are stochastic. Nevertheless, the qualitative results are consistent: the training with transformed returns results in better performance. With transformed returns, the agents sometimes get trapped in local optima with $F = 0$, which still results in significantly higher returns for the average agent.*

We can support these empirical findings by analyzing the optimal choice of $F$ under transformed and untransformed returns. We have already seen in equation 3 that without transformation, the expected return grows exponentially with time.

**Proposition 1.** *When solving the optimization problem in equation 2 under the dynamics in equation 7, the optimal $F \in [0, 1]$ is $F = 1$.*

*Proof.* When adding the parameter $F$, the dynamics in equation 3 change to

$$\mathbb{E}[R(T)] = R(0)(1 + 0.05F)^T.$$

Thus, $F$ should be as large as possible to maximize $\mathbb{E}[R(T)]$. □

Similarly, we can analyze the optimal choice of $F$ with transformed returns.

**Lemma 1.** *When solving the optimization problem in equation 2 for $\tilde{R}(T) = \log(R(T))$ under the dynamics in equation 7, the optimal $F \in [0, 1]$ is $F = 0.25$.*

*Proof.* For the expected value of logarithmic returns, we find (see also the appendix from Hulme et al. (2023))

$$\mathbb{E}[\log(R(T)] = \log(R(0)) + \frac{T}{2}(\log(1 + 0.5F) + \log(1 - 0.4F))$$
$$= \log(R(0)) + \frac{T}{2}\log(1 - 0.2F^2 + 0.1F).$$

We can maximize this by setting the derivative with respect to $F$ to 0:

$$\frac{T}{2}\frac{-0.4F + 0.1}{-0.2F^2 + 0.1F + 1} \overset{!}{=} 0,$$

from which we can find $F = 0.25$. $\qquad\square$

This result is also known as the Kelly criterion (Kelly, 1956).

In conclusion, when maximizing expected untransformed returns, we have an expected return of $100 \cdot 1.05^T$. However, the average agent ends up with a return of 0 almost surely as $T$ goes to infinity. For logarithmic returns, expected and time average growth rate coincide, as shown by Hulme et al. (2023). Then, the expected return at $T$ is

$$R(T) = R(0)\exp\left(\frac{T}{2}\log(1 - 0.2 \cdot 0.25^2 + 0.1 \cdot 0.25)\right) \approx 1.006^T R(0).$$

Thus, in expectation, the focus on expected return yields an $F$ with a higher growth rate ($1.05 > 1.006$). However, in the long run, optimizing logarithmic return yields a growth rate larger than one, while optimizing without transformation yields a return of 0 almost surely.

## 4 Learning an ergodicity transformation

In scenarios like the coin toss game, due to the perfect information of future returns, it is possible to derive a suitable transformation analytically—for a more detailed discussion, we refer the reader to Peters & Adamou (2018). However, the true power of reinforcement learning (RL) lies in its ability to handle complex environments for which we lack accurate analytical expressions. Therefore, it is desirable to learn transformations directly from data.

In this section, we develop an algorithm for learning such a transformation from data. In particular, we show how for a time-series of returns we can learn a transformation that transforms them into returns with increments that (approximately) follow equation 6. For the coin-toss game, where different policies amount to different choices of $F$, learning such a transformation once would be sufficient as the policy does not change the problem structure—choosing the logarithm always yields erogidc increments, no matter the choice of $F$. In the general RL setup, this might be different. We discuss how to embed the transformation into an RL algorithm that can handle also such cases in section 7.

The central characteristic of the transformation is captured by equation 6: we want the increments to be ergodic and, in particular, to be independent and identically distributed (i.i.d.) with constant variance. However, determining this i.i.d. property with a high degree of accuracy, especially from real-world data, can be challenging. Instead, we approximate the behavior of the transform to that of a variance-stabilizing transform.

**Definition 1** (Bartlett (1947)). *A* variance stabilizing transform *is defined as*

$$h(x) = \int\limits_0^x \frac{1}{\sqrt{v(u)}}\,\mathrm{d}u,$$

*with variance function $v(u)$ describing the variance of a random variable as a function of its mean.*

A variance stabilizing transform aims to transform a given time series into one with constant variance, independent of the mean (Bartlett, 1947). This is a generalization of our desired i.i.d. property as if the

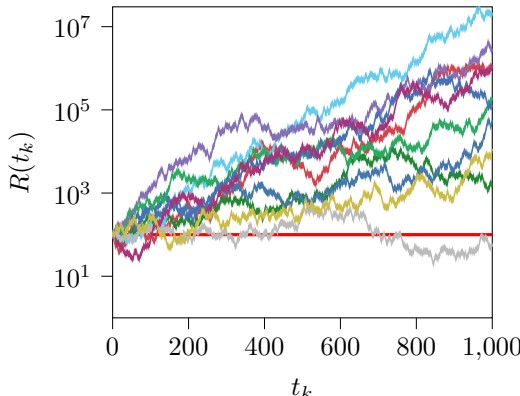

Figure 3: Learning bet strategies for the adapted coin toss game with learned transformation. *Similar to the logarithm, also with the learned transformation, the majority of the agents ends up winning.*

transformation $h(R(t_k))$ has i.i.d. increments, then the increments also have constant variance, independent of the mean. In particular, it reflects what we define in equation 6, where the mean is determined by the drift $\mu$, and we have defined that the random variable $v(t_k)$ must have finite variance. The i.i.d. property implied by equation 6 might then not be achieved. However, we show later in this section as well as in section 7 that also with this approximation the transformation is effective. Thus, our objective becomes finding a variance stabilizing transform following definition 1. In our case, as we want to stabilize the variance of the increments, we adapt the original definition of the variance function $v(u)$ in definition 1 to

$$v(u) = \text{Var}[R(t_{k+1}) - R(t_k) \mid R(t_k) = u].$$

This variance function represents the variance of the following increment as a function of the current transformed return.

The approach for estimating $v(u)$ from data is inspired by the additivity and variance stabilization method for regression (Tibshirani, 1988). Estimating $v(u)$ first involves plotting $R(t_k)$ against $\log((R(t_{k+1}) - R(t_k) - \hat{\mu})^2)$, with $\hat{\mu}$ the empirical mean of the increments. In our setting, the mean of the increments of the original untransformed process may not be constant throughout a trajectory. Hence, assuming a constant $\hat{\mu}$ results in small values having an over-estimated variance and large values having an under-estimated variance. The straightforward way to fix this would be to estimate $\mu(u)$ as a function of $u$; however, this introduces a further estimation problem. Instead, we can estimate the second-moment function and use this as a proxy for the variance function,

$$\mu^2(u) = \mathbb{E}[(R(t_{k+1}) - R(t_k))^2 \mid R(t_k) = u].$$

In appendix A.2, we show that $\mu^2(u) \propto v(u)$, which is satisfactory for our needs as if the process $R(t_k)$ has i.i.d. increments, then so will the process $a \cdot R(t_k)$ for any $a \in \mathbb{R}$.

To estimate the function $\log(\mu^2(u))$ we plot $R(t_k)$ against $\log((R(t_{k+1}) - R(t_k))^2)$. Then, fitting a curve represents taking the expected value. We use the locally estimated scatter-plot smoothing (LOESS) method (Cleveland, 1979). The reason behind estimating $\log(\mu^2(u))$ is that this guarantees $\mu^2(u)$ always to be positive, which is vital as the variance stabilizing transform requires us to take the square root. This approach follows the reasoning by Tibshirani (1988).

Having derived this transformation, we apply it to the coin toss game. We first collect a return trajectory with $F = 1$. Based on this trajectory, we learn an ergodicity transformation following the steps described in this section. Then, we again train a PPO agent but feed it the increments of transformed returns as previously with the logarithmic transformation. As before, we execute the learned policy 100 times for 1000 time steps each and show rollouts for the first ten agents in figure 3. Also with this transformation, most agents end up learning winning strategies. The statistics confirm this: across all 100 agents, we have a median return of around $17\,517$ and an average return of around $956\,884$. Thus, we conclude that we can learn a suitable transformation from data, enabling PPO to learn a policy that benefits individual agents in the long run.

We provide a Python implementation of the transformation and the coin toss example in the supplementary material.

## 5   Risk-sensitive RL

The ergodicity transformation serves as a means for RL agents to optimize the long-term performance of individual returns, enabling the learning of robust policies, as demonstrated in figure 3. Another approach to improving the robustness of RL algorithms is through risk-sensitive RL. While risk-sensitive RL is not motivated by ergodicity, it also proposes transforming returns. Inspired by Peters & Adamou (2018), we can analyze these transformations and determine under which dynamics they yield transformed returns with ergodic increments. This analysis allows us to gain insights into which type of transformation may offer robust performance in which settings.

Here, we focus on the exponential transformation,

$$h_{\mathrm{rs}}(R) \coloneqq \beta \exp(\beta R),$$

where $\beta \in \mathbb{R} \setminus \{0\}$ is a hyperparameter with $\beta < 0$ the "risk-averse", and $\beta > 0$ "risk-seeking" case. If this transformation were an ergodicity transformation, then its increments $h_{\mathrm{rs}}(R(t_k)) - h_{\mathrm{rs}}(R(t_k - 1))$ would follow equation 6. If we now assume that the dynamics of the return $R(t_k)$ belong to the class of Itô processes, i.e., a general class of stochastic processes, we can derive a concrete equation describing the return dynamics. This derivation becomes relatively technical, and we defer it to the appendix (appendix A.3). Here, we only present the result and discuss its implications. We can derive the return dynamics as

$$R_t = \frac{1}{\beta} \ln \left| \frac{\sigma}{\beta} \right| + \frac{1}{\beta} \ln \left| \frac{\mu}{\sigma} t + W_t + \frac{\beta}{\sigma} \right|. \tag{8}$$

The obtained return dynamics are logarithmic in time. Logarithmic returns (or regrets) are common in the RL literature (He et al., 2021; Velegkas et al., 2022; Yang et al., 2021). Consider a scenario where a robot arm must reach a set point, and the reward is defined as the negative distance to that set point. Initially, rapid progress can be made by moving quickly in the roughly correct direction. As the robot gets closer, the movement becomes more fine-grained and slower, resulting in slower progress. By using an exponential transformation, we counteract this phenomenon, ensuring that all time steps contribute equally to the return.

We next apply the exponential transformation for various choices of $\beta$ to the coin-toss game and test both the "risk-averse" and the "risk-seeking" setting. For the risk-seeking setting ($\beta > 0$), we quickly run into numerical problems. The coin-toss problem has itself exponential dynamics, and thus, returns can get large. Exponentiating those again lets us reach the limits of machine precision. For the risk-averse setting ($\beta < 0$), we consistently learn constant policies with $F = 0$. While this is still better than the policies standard PPO learned, it cannot compete with the results from figure 3.

This outcome is not surprising. From an ergodicity perspective, the exponential transformation is only suitable if the dynamics are logarithmic. The dynamics of the coin-toss game are exponential, which is precisely the inverse behavior. Thus, we would not expect the transformation to yield good policies, as is confirmed by our experiments.

## 6   Ergodicity in RL and related work

In this section, we first discuss how our definition of ergodicity relates to the more standard one typically considered in RL. We then discuss the significance of non-ergodicity in RL, relate our contributions to the RL literature, and briefly discuss connections to economics.

### 6.1   Ergodic rewards vs. ergodic MDPs

When (non-)ergodicity is discussed in the RL literature, this notion is typically related to the underlying MDP; see, for instance, Chapter 10 by Sutton & Barto (2018) or Chapter 8 by Puterman (2014). If an MDP

is ergodic, we converge to a stationary state distribution on which the initial state has no influence. Based on this definition, there has been work within the RL community that provides guarantees while explicitly assuming ergodicity (Pesquerel & Maillard, 2022; Ok et al., 2018; Agarwal et al., 2022) or by guaranteeing to avoid any states within an "absorbing" barrier, i.e., only exploring an ergodic sub-MDP (Turchetta et al., 2016; Heim et al., 2020). For Q-learning, Majeed & Hutter (2018) has shown convergence even for non-ergodic and non-MDP processes. Nevertheless, none of these works, as a consequence of non-ergodicity, question the use of the expectation operator in the objective function.

In general, the concepts of ergodic MDPs and ergodic rewards are related, although not identical. For instance, consider a trivial MDP where we stay in the initial state with probability 1 and, in every state, receive a reward of +1 in every time step. Clearly, the MDP is non-ergodic, as the initial state is not forgotten, but the expected value and time average of rewards are identical. Conversely, consider a two-state MDP with a transition probability of 50 % between the two states. Clearly, the state distribution is stationary, and the initial state has no influence. Consider further that we receive a reward of +1 in state 1 and a reward of -1 in state 2. This is an unbiased random walk. The expected reward for every step $t_k$ is 0. However, the return for $T \to \infty$ is a random variable with unbounded variance. Thus, equation 5 does not hold almost surely.

Generally, an ergodic MDP does result in a stationary state distribution. The reward function is, typically, a deterministic function from the state (and potentially action) space to the real numbers. Thus, also the reward distribution is stationary. While an ergodic stochastic process is stationary, the converse does not necessarily hold. The conditions under which stationary processes are ergodic have, for instance, been discussed by Paŕzen (1958).

## 6.2 Significance of non-ergodic rewards

The coin-toss game is an excellent example to illustrate the problem of maximizing the expected value of non-ergodic rewards. When maximizing non-ergodic rewards, we may end up with a policy that receives an arbitrarily high return with a probability that tends to zero in time but leads to failure almost surely. Also in less extreme cases, the expected value prefers risky policies if their return in case of success outweighs the failure cases. This results in learning non-robust policies, a behavior frequently observed in standard RL algorithms (Amodei et al., 2016; Leike et al., 2017; Russell et al., 2015).

Non-ergodicity is not unique to the coin-toss game. Peters & Klein (2013) have shown that geometric Brownian motion (GBM) is a non-ergodic stochastic process. GBM is commonly used to model economic processes, a domain where RL algorithms are increasingly applied (Charpentier et al., 2021; Zheng et al., 2022). Thus, especially in economics, ergodicity should not simply be assumed. Nevertheless, the example of GBM is also informative for other applications. Generally, RL is most interesting when the environment dynamics are too complex to model, i.e., we usually deal with nonlinear dynamics. If already a linear stochastic process such as GBM is non-ergodic, we cannot assume ergodicity for the general dynamics we typically consider in RL. This is confirmed by the amount of recent research on RL with heavy-tailed rewards (Zhuang & Sui, 2021; Huang et al., 2024; Zhu et al., 2024), suggesting that it is an important problem.

Another way of "ergodicity-breaking" is often motivated using the example of Russian roulette (Ornstein, 1973). When multiple people play Russian roulette for one round each, and their average outcome is considered, the probability of death is one in six. However, if a single person plays the game infinitely many times, that person will eventually die with probability one. In the context of RL, this is akin to the presence of absorbing barriers or safety thresholds that an agent must not cross. Particularly in RL applications where the consequences of failure can be catastrophic, such as in autonomous driving (Brunke et al., 2022), these safety thresholds become vital.

How big the impact of non-ergodic rewards is and, hence, what improvement we can expect from the transformation we propose in section 4 depends on "how non-ergodic" the rewards are. The physics community has introduced different measures for how much a process deviates from being ergodic (Scott et al., 2009; Mathew & Mezić, 2011). The larger the difference between the time average and expected value, the higher the impact of optimizing transformed returns.

### 6.3 Related work

In this paper, we propose to transform returns to deal with non-ergodic rewards. In the previous section, we have shown how a popular transformation from risk-sensitive RL (Mihatsch & Neuneier, 2002; Shen et al., 2014; Fei et al., 2021; Noorani & Baras, 2021; Noorani et al., 2022; Prashanth et al., 2022) can be motivated from an ergodicity perspective. Reward-weighted regression (Peters & Schaal, 2007; 2008; Wierstra et al., 2008; Abdolmaleki et al., 2018; Peng et al., 2019) also proposes to use transformations, but the transformations are typically justified using intuitive arguments instead of from an ergodicity perspective. Interestingly, most existing work also uses an exponential transformation, which is the cornerstone of risk-sensitive control. Thus, the analysis we have done for risk-sensitive RL also applies to reward-weighted regression. In the risk-sensitive RL literature, we also find works different transformations than the exponential one. For instance, Prashanth & Ghavamzadeh (2013) estimate the variance of returns and incorporate it into a transformation, Tamar et al. (2015) provide a more general analysis of different risk measures. Investigating also such approaches from an ergodicity perspective would be an interesting study for future work.

Another approach that optimizes transformed returns is Bayesian optimization for iterative learning (BOIL) (Nguyen et al., 2020). BOIL is developed for hyperparameter optimization. While this setting differs from the one we consider, we show in appendix A.4.1 that the transformation used in BOIL can be replaced with ours, leading to similar or better results.

Through the ergodicity transformation, we seek to optimize the long-term performance of RL agents. Improving the long-term performance of RL agents in continuous tasks is also the goal of average reward RL. The idea of optimizing the average reward criterion originated in dynamic programming (Howard, 1960; Blackwell, 1962; Veinott, 1966), and has already in the early days of RL been taken up to develop various algorithms, see, for instance, the survey by Mahadevan (1996). Also in recent years, the average reward criterion has been used for novel RL algorithms (Zhang & Ross, 2021; Wei et al., 2020; 2022). In average reward RL, we still take the expected value of the reward function *and* let time go to infinity. Were the reward function ergodic, it would not matter whether we first take the expected value or first let time go to infinity. However, for a non-ergodic function, it does. In average reward RL, we first take the expected value. For the coin-toss game, that would yield an optimization criterion that grows exponentially while the set of agents that obtain a return larger than zero shrinks to a set of measure zero as time goes to infinity. Thus, average reward RL may fall into the same trap as conventional RL when dealing with non-ergodic reward functions.

A further research direction in RL to which our approach can be related is reward shaping (Ng et al., 1999; Tang et al., 2017; Zheng et al., 2018; Memarian et al., 2021). In reward shaping, we typically try to adapt the existing reward function, for instance, to deal with sparse rewards or to encourage exploration. Usually, the new reward function is then a summation of the original reward and a new element. This new element introduced by reward-shaping techniques can be designed based on prior knowledge about the environment or learned from data. In our case, instead of adding a new element to an existing reward function, we transform the entire return. Thus, our approach fundamentally differs from existing reward-shaping techniques. However, the result of the transformation is that the increments of transformed returns follow equation 6. Thus, we have *linear*, stochastic return dynamics. Such dynamics should be easier to optimize through gradient-based RL algorithms than the arbitrary dynamics that we typically assume. Therefore, reward-shaping may serve as an additional motivation for our approach.

### 6.4 Connections to economics

The transformations used in risk-sensitive RL are often motivated by economics, in particular, by utility (Bernoulli, 1954) and prospect (Kahneman & Tversky, 1997) theory. These theories rely on psychological arguments to argue that some humans are more "risk-averse" than others. Peters & Adamou (2018) have shown how acknowledging non-ergodicity and that humans are more likely to optimize the long-term return than an average over an ensemble of infinitely many trajectories can recover widespread transformations used in utility theory. Empirical research (Meder et al., 2021; Vanhoyweghen et al., 2022; Skjold et al., 2024) has further shown that this treatment can better predict actual human behavior. The ergodicity perspective does not rely on psychology as an explanation; instead, it explains psychological observations. It is, in this sense,

more fundamental and, as a result, more general, namely applicable to cases where psychology cannot be invoked, particularly to inanimate optimizers such as machines devoid of a psyche.

# 7 Proof-of-concept

The coin-toss game, while illustrative, represents a simplified scenario. As already mentioned in section 4, in coin-toss game, learning a single transformation is sufficient as the policy parameter $F$ has no influence on the problem structure. In RL, this is not necessarily the case. Thus, we here embed the transformation into an RL algorithm. In particular, we here develop an ergodic REINFROCE algorithm as an example for Monte Carlo-based algorithms. That is, REINFORCE always collects a return trajectory and then uses this trajectory to update its weights. In our setting, this is advantageous as it allows us to learn a transformation using the collected trajectory. We summarize ergodic REINFORCE in algorithm 1.

---

**Algorithm 1** Pseudocode of the ergodic Monte Carlo-based RL algorithm.

---
1: **for** $i$ in num_episodes **do**
2:      $R(t_0), \ldots, R(r_K) \leftarrow \text{rollout}(\pi_i)$
3:      $\log(\mu_{\pi_i}^2(t_k)) \leftarrow \text{LOESS}(\log((R(t_{k+1}) - R(t_k))^2)$
4:      $h(t_k) \leftarrow \int_0^K \exp(\mu_{\pi_i}^2(\kappa)) \, \mathrm{d}\kappa$
5:      $\tilde{r} \leftarrow \frac{1}{K} \sum_{\kappa=1}^{K} (h(R(t_\kappa)) - h(R(t_{\kappa-1})))$
6:      $\mu_{i+1} \leftarrow \text{update\_policy}(\tilde{r})$

---

We evaluate ergodic REINFORCE on two classical RL benchmarks: the cart-pole system and the reacher, using the implementations provided by Brockman et al. (2016). In both cases, we seek to evaluate whether or not the ergodicity transformation increases robustness. James et al. (1994) have established that robustness to heavy-tail distributions is related to robustness against model uncertainties. Thus, we evaluate robustness by changing model parameters between training and testing. Specifically, we change the length of the pole for the cart-pole system from 0.5 to 1 and the length of one of the links for the reacher from 1 to 1.5. Besides, the cart-pole system has a non-ergodic reward structure as there exists an absorbing state: the episode ends if the pole drops.

We compare the ergodic REINFORCE algorithm that is trained as shown in algorithm 1 with the standard REINFORCE algorithm trained with untransformed rewards. In figure 4, we show the mean and standard deviation of the return over five runs. For both algorithms, we plot the untransformed returns. Further details on hyperparameter choices are provided in appendix A.5.

**Cart-pole.** In the cart-pole environment, the objective is to maintain the pole in an upright position for as long as possible. To evaluate the long-term performance of the ergodicity transformation, we train the algorithm using episode lengths of 100 time steps but test it with episodes lasting 200 time steps. Thus, as we see in figure 4a, the return during testing is higher than during training. We can also see that for ergodic REINFORCE, the agent is closer to the optimal reward of 200 during testing. The standard REINFORCE algorithm performs slightly worse. Thus, we can see that leveraging the ergodicity transformation improves the long-term performance compared to the standard algorithm.

**Reacher.** In the reacher environment, we aim to track a set point with the end of the last link. Thus, extending the episode length does not make sense in this setting. However, this is unnecessary to demonstrate the advantage of using the ergodicity transformation. In figure 4b, we see that, while both algorithms successfully improve their return during training, ergodic REINFORCE even more than standard REINFORCE, ergodic REINFORCE can generalize to the new link length and mass during testing. Standard REINFORCE ends up with close to minimal reward during testing.

# 8 Conclusions and limitations

This paper discussed the impact of ergodicity on the choice of the optimization criterion in RL. If the rewards are non-ergodic, focusing on the expected return yields non-robust policies that we currently find with

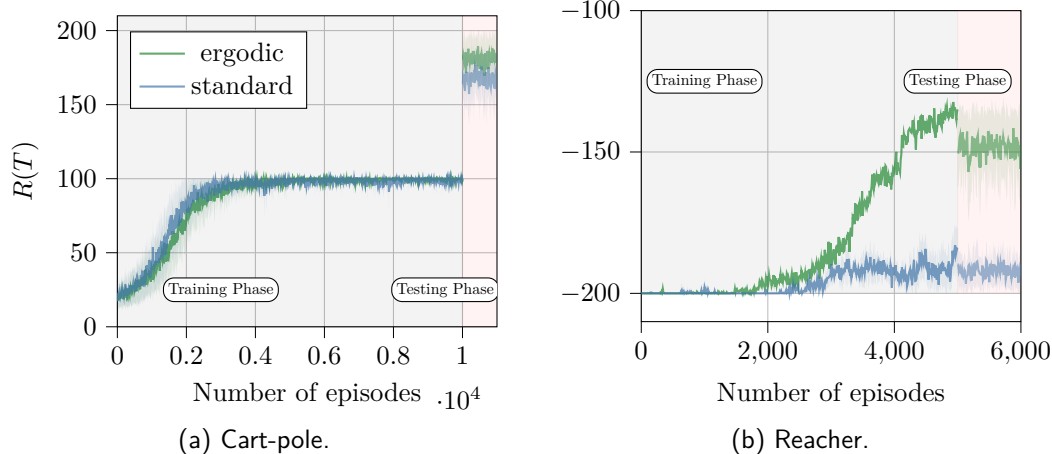

Figure 4: Ergodic vs. standard REINFORCE on common benchmarks. *For the cart-pole, we see slight improvements when using the ergodicity transformation, while for the reacher, only ergodic REINFORCE learns a successful policy.*

conventional RL algorithms. An alternative to changing the objective and, with this, having to come up with entirely new RL algorithms is trying to find an ergodicity transformation. We presented a method for learning an ergodicity transformation that converts a time series of returns into a time series with ergodic increments. Then, optimizing the expected value of those ergodic increments is equivalent to maximizing the long-term growth rate of the return. We further showed how the ergodicity perspective provides a theoretical foundation for transformations used in risk-sensitive RL. We demonstrated the effectiveness of the proposed transformation on standard RL benchmark environments.

This paper is the first step toward acknowledging non-ergodicity of reward functions and focusing on the long-term return and, with that, robustness in RL. This opens various directions for future research. Firstly, addressing the challenge of transforming returns in RL algorithms that update weights incrementally rather than relying on episodic data remains an open question. Secondly, our transformation currently focuses solely on the current return, but returns may also depend on the current state of the system, suggesting the possibility of state-dependent transformations. Then, also investigating the computational complexity and trading off potentially more robust performance with the additional complexity through the transformation is a crucial aspect. Thirdly, extending this research to multi-agent RL could be promising, building on insights by Fant et al. (2023) and Peters & Adamou (2022) regarding the impact of non-ergodicity on the emergence of cooperation in biological multi-agent systems. Finally, investigating the connection between optimizing time-average growth rates instead of expected values and discount factors, as explored by Adamou et al. (2021), could make the discount factor as a hyperparameter in RL dispensable.

## Acknowledgements

This research was partially supported by the project *NewLEADS - New Directions in Learning Dynamical Systems* (contract number: 621-2016-06079), funded by the Swedish Research Council and by *Kjell och Märta Beijer Foundation*, and by the Engineering and Physical Sciences Research Council through the Mathematics of Systems II Centre for Doctoral Training at the University of Warwick (reference EP/S022244/1).

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

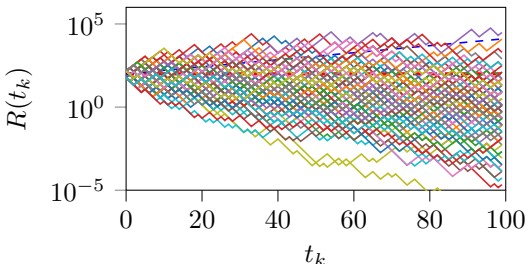

Figure 5: Coin-toss with fewer iterations. *With fewer iterations and more trajectories, we see that a few do end up with a higher than the initial return.*

# A Appendix

## A.1 Coin-toss example with fewer iterations

In figure 1a, we have seen that while the expected return is increasing, all individual trajectories end up with a lower than the initial return. The reason is, as shown in figure 1b, that the number of sample paths that end up "winning" decreases as the trajectory length increases. To verify that there are actually sample paths that end up with a higher return, we show in figure 5 another simulation of the coin-toss game. We here simulate the game for only 100 iterations to increase the probability of positive sample paths and simulate 100 independent games. We see that there are indeed sample paths where the final return is higher than 100. Nevertheless, those are a minority, and in most cases, we still end up with a lower return.

## A.2 Proportionality of variance and second moment functions

The variance-stabilizing transform $h(x)$ is unique up to linear transformations. That is, the function $ah(x) + b$ for $a \in \mathbb{R}^+$, $b \in \mathbb{R}$ will also produce a time series with the desired properties. Thus, we only need to estimate the variance function up to a scalar multiplier. In the following, we approximate $\mu^2(u) := \mathbb{E}[(R(t_{k+1}) - R(t_k))^2 \mid R(t_k) = u]$ using a Taylor series expansion and show that $v(u) \propto \mu^2(u)$. In particular, we have

$$
\begin{aligned}
\mu^2(u) &= \mathbb{E}[(R(t_{k+1}) - R(t_k))^2 \mid R(t_k) = u] \\
&= \mathbb{E}[R(t_{k+1})^2 \mid R(t_k) = u] \\
&\quad - 2\mathbb{E}[R(t_{k+1}) \mid R(t_k) = u][R(t_k) \mid R(t_k) = u] \\
&\quad + \mathbb{E}[R(t_k)^2 \mid R(t_k) = u] \\
&= \mathbb{E}[R(t_{k+1})^2 \mid R(t_k) = u] \\
&\quad - 2u\mathbb{R}[R(t_{k+1}) \mid R(t_k) = u] + u^2.
\end{aligned}
\tag{9}
$$

We now perform a second-order Taylor expansion with the function $h^{-1}$ on the random variable $h(R(t_{k+1}))$ to find $\mathbb{E}[R(t_{k+1}) \mid R(t_k) = u]$,

$$
\begin{aligned}
&\mathbb{E}[R(t_{k+1}) \mid R(t_k) = u] \\
={}&\mathbb{E}[h^{-1}(h(R(t_{k+1}))) \mid R(t_k) = u] \\
={}&\mathbb{E}[h^{-1}(h(u) + h(R(t_{k+1})) - h(u)) \mid R(t_k) = u] \\
\simeq{}&\mathbb{E}[h^{-1}(h(u)) + (h^{-1})'(h(u))(h(R(t_{k+1})) - h(u)) \\
&+ \frac{1}{2}(h^{-1})''(h(u))(h(R(t_{k+1})) - h(u))^2 \mid R(t_k) = u] \\
={}&m_1(h^{-1})'(h(u)) + \frac{m_2}{2}(h^{-1})''(h(u)).
\end{aligned}
$$

In the final step, as $h(u)$ is a function that transforms the original time series into a time series with independent increments, we can assume that, for all $n \in \mathbb{N}$,

$$\mathbb{E}[(h(R(t_{k+1})) - h(u))^n \mid R(t_k) = u] = m_n \in \mathbb{R}.$$

That is, the moments of the transformed increments are stationary over the state space. We can then use the inverse-function rule to calculate the derivatives as

$$(h^{-1})'(h(u)) = \frac{1}{h'(h^{-1}(h(u)))} = \frac{1}{h'(u)}$$

$$(h^{-1})''(h(u)) = \frac{-h''(h^{-1}(h(u)))}{h'(h^{-1}(h(u)))^3} = \frac{-h''(u)}{h'(u)^3}.$$

Hence, we have

$$\mathbb{E}[R(t_{k+1})^2 \mid R(t_k) = u] \simeq u + \frac{m_1}{h'(u)} - \frac{m_2 h''(u)}{2h'(u)^3}.$$

We use a similar method to find $\mathbb{E}[R(t_{k+1})^2 \mid R(t_k) = u]$. However, this time, we perform the Taylor expansion with the squared-inverse function $h^{-2}(x) := (h^{-1}(x))^2$,

$$\begin{aligned}
&\mathbb{E}[R(t_{k+1})^2 \mid R(t_k) = u] \\
&= \mathbb{E}[h^{-2}(h(R(t_{k+1}))) \mid R(t_k) = u] \\
&\simeq u^2 + m_1(h^{-2})'(h(u)) + \frac{m_2}{2}(h^{-2})''(h(u)).
\end{aligned}$$

We can use the chain rule to calculate the derivatives of the squared-inverse function,

$$(h^{-2})'(h(u)) = 2(h^{-1})'(h(u))h^{-1}(h(u))\frac{2u}{h'(u)}$$

and

$$\begin{aligned}
(h^{-2})''(h(u)) &= 2((h^{-1})''(h(u))h^{-1}(h(u)) + (h^{-1})'(h(u))^2) \\
&= \frac{-2uh''(u)}{h'(u)^3} + \frac{2}{h'(u)^2}.
\end{aligned}$$

Hence, we have

$$\mathbb{E}[R(t_{k+1})^2 \mid R(t_k) = u] \simeq u^2 + \frac{2um_1}{h'(u)} - \frac{m_2 u h''(u)}{h'(u)^3} + \frac{m_2}{h'(u)^2}.$$

Substituting into equation 9 gives us

$$\begin{aligned}
\mu^2(u) &= \mathbb{E}[R(t_{k+1})^2 \mid R(t_k) = u] \\
&\quad - 2u\mathbb{E}[R(t_{k+1}) \mid R(t_k) = u] + u^2 \\
&\simeq \left(u^2 + \frac{2um_1}{h'(u)} - \frac{m_2 u h''(u)}{h'(u)^3} + \frac{m_2}{h'(u)^2}\right) \\
&\quad - 2u\left(u + \frac{m_1}{h'(u)} - \frac{m_2 h''(u)}{2h'(u)^3}\right) + u^2 \\
&= \frac{m_2}{h'(u)^2}.
\end{aligned}$$

Finally, we can use the fundamental theorem of calculus on definition 1 to get

$$v(u) = \frac{1}{h'(u)^2} \implies \mu^2(u) \propto v(u) \text{ (approximately)}.$$

### A.3 Derivation of the risk-sensitive reward function equation 8

For the sake of clarity, we perform our analysis in continuous time. We assume that the return follows an arbitrary Itô process

$$\mathrm{d}R = f(R)\,\mathrm{d}t + g(R)\,\mathrm{d}W(t), \tag{10}$$

where $f(R)$ and $g(R)$ are arbitrary functions of $R$ and $W(t)$ is a Wiener process. This captures a large class of stochastic processes, as both $f$ and $g$ can be nonlinear and even stochastic. Assume now that the risk-sensitive transformation $h_{\mathrm{rs}}$ extracts an ergodic observable from equation 10. Then, its increments follow a Brownian motion, i.e., the continuous-time version of equation 6:

$$\mathrm{d}h_{\mathrm{rs}} = \mu\,\mathrm{d}t + \sigma\,\mathrm{d}W(t). \tag{11}$$

As we know $h_{\mathrm{rs}}$, we now seek to find $f$ and $g$ for which equation 11 holds.

Following Itô's lemma (Itô, 1944), we can write $\mathrm{d}R$ as

$$\mathrm{d}R = \left(\frac{\partial R}{\partial t} + \mu\frac{\partial R}{\partial h_{\mathrm{rs}}} + \frac{1}{2}\sigma^2\frac{\partial^2 R}{\partial h_{\mathrm{rs}}^2}\right)\mathrm{d}t + \sigma\frac{\partial R}{\partial h_{\mathrm{rs}}}\,\mathrm{d}W(t). \tag{12}$$

As we can invert $h_{\mathrm{rs}}(R)$ such that $R(h_{\mathrm{rs}}) = \frac{\ln\left(\frac{h_{\mathrm{rs}}}{\beta}\right)}{\beta}$ and since the inverse is twice differentiable, we can insert it into equation 12 and obtain

$$\begin{aligned}
\mathrm{d}R &= \left(\frac{\mu}{\beta h_{\mathrm{rs}}} - \frac{1}{2}\frac{\sigma^2}{\beta h_{\mathrm{rs}}^2}\right)\mathrm{d}t + \frac{\sigma}{\beta h_{\mathrm{rs}}}\,\mathrm{d}W(t) \\
&= \left(\frac{\mu}{\beta^2\exp(\beta R)} - \frac{1}{2}\frac{\sigma^2}{\beta^3\exp(2\beta R)}\right)\mathrm{d}t \\
&\quad + \frac{\sigma}{\beta^2\exp(\beta R)}\,\mathrm{d}W(t).
\end{aligned} \tag{13}$$

This equation provides valuable insights into the role of $\beta$. Specifically, it highlights that the volatility term (the coefficient of $\mathrm{d}W(t)$) is always positive, regardless of the sign of $\beta$. However, the drift term (the coefficient of $\mathrm{d}t$) depends on the sign of $\beta$. For $\beta < 0$, the drift term is positive, while for $\beta > 0$, it starts negative when $\beta$ is small and then turns positive as $\beta$ increases.

From an ergodicity perspective, the risk-averse variant with $\beta < 0$ is suitable when equation 13 exhibits a positive drift, while the risk-seeking variant with $\beta > 0$ is more appropriate when equation 13 has a negative drift. This aligns with intuitive reasoning: when the drift is negative, there is limited gain from caution, and one might choose to go all in and hope for luck. This is also the case when the drift is too small to outweigh the volatility.

The differential dynamics in equation 13 have a closed-form solution. We start the derivations by simplifying equation 13. We introduce $k(R) := \frac{\sigma}{\beta^2\exp(\beta R)}$ and $c_{\mathrm{v}} := \frac{\sigma}{\mu}$, which results in

$$\mathrm{d}R = \left(\frac{1}{c_{\mathrm{v}}}k(R) + \frac{1}{2}k(R)k'(R)\right)\mathrm{d}t + k(R)\,\mathrm{d}W(t).$$

From this, we can see that the resulting stochastic differentiable equation belongs to the class of reducible SDEs and has a known, general solution (Kloeden & Platen, 1992, pp. 123–124):

$$R_t = \ell^{-1}\left(\frac{1}{c_{\mathrm{v}}}t + W_t + l(0)\right),$$

where $\ell(r) := \int^r \frac{ds}{k(s)} = \int^r \frac{\beta^2}{\sigma}\exp(\beta s)ds$. Now, we need to find an expression for $\ell(R)$:

$$\ell(R) = \int^R \frac{\beta^2}{\sigma}\exp(\beta s)\,\mathrm{d}s = \frac{\beta}{\sigma}\exp(\beta R).$$

This expression is invertible,

$$\ell^{-1}(R) = \frac{1}{\beta} \ln\left|\frac{\sigma}{\beta}\right| + \frac{1}{\beta} \ln|R|.$$

Thus, we finally obtain equation 8:

$$R_t = \ell^{-1}\left(\frac{1}{c_{\mathrm{v}}}t + W_t + l(0)\right) = \ell^{-1}\left(\frac{\mu}{\sigma}t + W_t + \frac{\beta}{\sigma}\right)$$
$$= \frac{1}{\beta}\ln\left|\frac{\sigma}{\beta}\right| + \frac{1}{\beta}\ln\left|\frac{\mu}{\sigma}t + W_t + \frac{\beta}{\sigma}\right|.$$

### A.4  Hyperparameter optimization

Besides the experiments presented in the main body, we also compared our learned transformation in a hyperparameter optimization task with the BOIL (Bayesian optimization for iterative learning) algorithm (Nguyen et al., 2020). Before presenting the results, we briefly introduce BOIL.

#### A.4.1  Bayesian optimization for iterative learning

Boil aims to train a machine learning algorithm given a $d$-dimensional hyperparameter $x \in \mathcal{X} \subset \mathbb{R}^d$ for $T$ iterations. This process produces training evaluations $R(\cdot \mid x, T)$ with $T \in [T_{\min}, T_{\max}]$. These evaluations could generally be returns of an episode in RL or training accuracies in deep learning. Here, we focus on episode returns in reinforcement learning. Given the raw training curve $R(\cdot \mid x, T)$, BOIL assumes an underlying, smoothed black-box function $f$ and then aims to find $x^* = \arg\max_{x \in \mathcal{X}} f(x, T_{\max})$. This black-box function is modeled as a Gaussian process (GP), and the next set of hyperparameters is selected using a variation (Wang & de Freitas, 2014) of the expected improvement (Jones et al., 1998) algorithm.

Existing Bayesian optimization approaches for hyperparameter optimization typically define the objective function as an average loss over the final learning episodes. This ignores how stable the performance is and might be misleading due to the noise and fluctuations of observed episode returns, especially during early stages of training. Therefore, in BOIL, the authors propose compressing the whole learning curve into a numeric score via a preference function. In particular, they use the Sigmoid function (specifically, the Logistic function) to compute this "utility score" as

$$y = \hat{y}(R, m_0, g_0) = R(\cdot \mid, x, T) \cdot \ell(\cdot \mid m_0, g_0)$$
$$= \sum_{u=1}^{t} \frac{R(u \mid x, T)}{1 + \exp(-g_0(u - m_0))}, \tag{14}$$

where $\cdot$ is a dot product, and the Logistic function $\ell(\cdot \mid m_0, g_0)$ is parameterized by a growth parameter $g_0$ defining the slope and the middle point of the curve $m_0$. The choice of the Sigmoid function is mainly motivated by intuitive arguments. Since early weights are small, less credit is given to fluctuations at the initial stages, making it less likely for the surrogate function to be biased toward randomly well-performing settings. As weights monotonically increase, hyperparameters with improving performance are preferred. As weights saturate, stable, high-performing configurations are preferred over short "performance spikes" which often characterize unstable training. The score assigns higher values to the same performance if it is being maintained over more episodes.

The intuition provided by Nguyen et al. (2020) is that the optimal parameters $m_0, g_0$ will lead to a better fit of the GP, resulting in better prediction and optimization performance. The authors then parameterize the GP log marginal likelihood in terms of $m_0$ and $g_0$ and optimize both parameters using multi-start gradient descent.

#### A.4.2  Comparison

We tried to apply BOIL to the coin toss game, i.e., we tried to optimize hyperparameters for an RL algorithm on the coin toss game using BOIL. Unfortunately, we there ran into numerical problems caused by the large

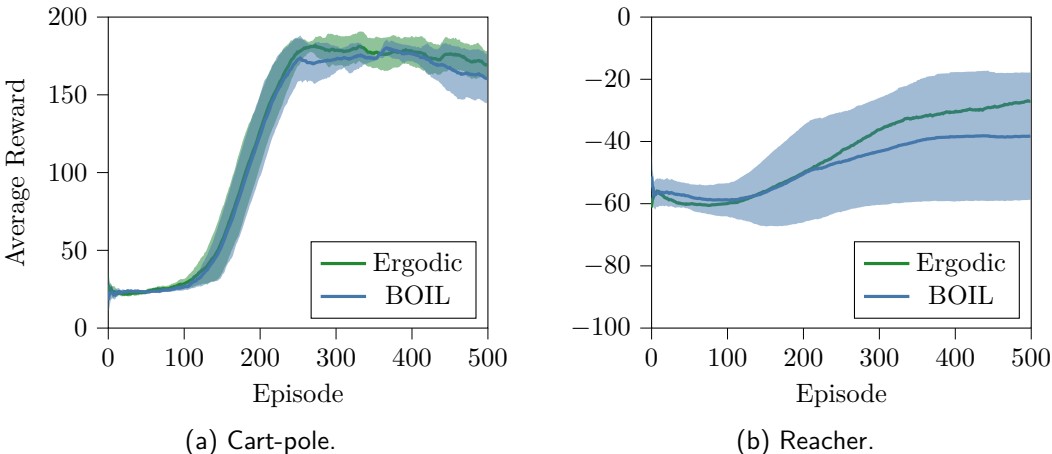

Figure 6: Comparison of BOIL and our transformation for hyperparameter optimization of deep RL algorithms. *Our non-parametric transformation performs at least en par with state-of-the-art hyperparameter optimization algorithms.*

values the return can have in some runs. Therefore, we compare BOIL to our learned transformation on the same benchmarks as we used in section 7 as they were also used by Nguyen et al. (2020). However, instead of learning policies, we optimize hyperparameters of deep RL algorithms that try to learn those policies. This is slightly different from the setting we designed our transformation for, and in that sense, it also challenges its generality. The used deep RL algorithms are the double deep Q-networks (DDQN) (Van Hasselt et al., 2016) algorithm for the cart-pole and the advantage actor-critic (A2C) algorithm (Mnih et al., 2016) for the reacher. In both cases, we tune the learning rate(s) and the discount factor. We adopt the code from Nguyen et al. (2020), only adding the ergodicity transformation but without changing any parameter settings.

We show the mean and standard deviation of the average return over five training runs in figure 6. The general, non-parametric transformation proposed in this paper achieves comparable performance as the tuned Sigmoid from Nguyen et al. (2020) on the cart-pole system and can outperform it on the reacher. This shows that while BOIL relies on intuitive arguments to develop a parametric transformation, we can achieve at least an en-par performance with a non-parametric transformation motivated from basic principles. Further, Nguyen et al. (2020) showed significant benefits of BOIL over existing hyperparameter optimization methods based on Bayesian optimization. Thus, coming up with reward transformations, in general, can significantly enhance learning. While the transformation in BOIL is designed for a specific setting, our transformation has a more universal character and is applicable in more diverse settings.

### A.5 Hyperparameter choices

The hyperparameter choices for the experiments in section 7 are provided in table 1.

Table 1: Hyperparameters for the experiments in section 7.

|                                   | Cart-pole | Reacher |
|-----------------------------------|-----------|---------|
| Discount rate                     | 0.99      | 0.99    |
| Training episodes                 | 1000      | 500     |
| Test episodes                     | 100       | 100     |
| Training episode length           | 100       | 200     |
| Test episode length               | 200       | 200     |
| Epochs                            | 10        | 10      |
| Nodes in the actor neural network | 16        | 64      |
| Learning rate                     | 0.0007    | 0.001   |

