# OpenReview forum: "Reinforcement learning with non-ergodic reward increments: robustness via ergodicity transformations"
_TMLR — Accepted by TMLR_

### Review · Reviewer_LhKg · 2024-11-09

**Summary Of Contributions:**

The paper addresses the challenge of non-ergodic reward processes in reinforcement learning (RL), highlighting that optimizing expected returns across many trajectories can lead to poor performance in individual runs. This issue is particularly critical in real-world applications like autonomous driving and financial trading, where robustness is essential.

To tackle this problem, the authors propose a novel method that transforms non-ergodic returns into ergodic ones using learned transformations. This approach allows standard RL algorithms to optimize long-term performance without modifying their core structures.

**Audience:**

Yes

**Broader Impact Concerns:**

None.

**Claims And Evidence:**

Yes

**Requested Changes:**

1. The example in Section 3 is intuitive and effective, but there appears to be some redundancy with the previous section's setup. Streamlining these sections could improve the paper's flow.
2. In Section 4, the authors approximate the transformation to behave like a variance-stabilizing transform. However, the implications of this approximation are not fully discussed. Elaborating on how this impacts the robustness of the learned policy would enhance understanding.
3. While the authors mention that ergodic Markov Decision Processes (MDPs) and ergodic rewards are related but defer formal analysis to future work, including a brief discussion on the relationship between these concepts in the current paper would be advantageous.

**Strengths And Weaknesses:**

**Strengths**

1. The focus on long-term performance over ensemble averages is highly relevant for deploying RL in high-stakes, real-world domains.
2. The identification and formalization of the non-ergodicity problem in RL are novel contributions.
3. The coin toss game example effectively illustrates the impact of non-ergodicity and the advantages of the proposed transformation.

**Weaknesses**

1. It's unclear how prevalent the heavy-tailed return distributions issue is in RL problems. A broader discussion on environments that naturally lead to long-tailed reward distributions would strengthen the paper. Additionally, exploring the relation to reward hacking—where agents exploit specific reward structures—could provide valuable context.
2. The potential connection to causal inference concepts, such as false correlations, also worth discussing. Agents might mistakenly associate certain actions with high rewards due to randomness rather than causality. Discussing this aspect could enhance the paper's impact.
3. The paper suggests that risk-sensitive RL methods typically rely on fixed transformations. If the proposed method generalizes existing approaches, the claim that risk-sensitive RL is limited to fixed transformations may not be entirely accurate. Clarification on how this approach extends or differs from traditional risk-sensitive methods would be beneficial.

---

> ### Author Response · Authors · 2024-11-25
>
> We thank the reviewer for the positive evaluation of our work. The significance of non-ergodic and heavy-tailed rewards is indeed an important topic. We discuss this in more detail in Section 6.2. We have there now also added some recent references (Zhuang & Sui, 2021; Huang et al., 2024; Zhu et al., 2024) studying explicitly heavy-tailed rewards to underscore the importance of this topic.
>
> Risk-sensitive RL indeed typically relies on fixed transformations. That is, a transformation is selected upfront and then applied during training. One popular example is the exponential transformation that we discuss in Section 5. Different from those approaches, we learn the transformation from data. In that sense, we generalize as the transformation fits the dynamic of the environment at hand.
>
> In Section 3, we introduce a slightly modified version of the coin-toss game as here, different from Section 2, we allow for an action $F$. Thus, we feel it necessary to spend some words on introducing the topic. However, if there are any concrete points where the reviewer feels the text is redundant, we are happy to streamline the section.
>
> In Equation (6), we formulate a strong requirement for the transformed increments. With the variance-stabilizing transform, we achieve that the variance of the increments is constant and independent of the mean, as desired by (6). However, we might not achieve the independence property implied by (6). We see empirically in Section 4 and Section 6 that even with this shortcoming, the transformation does a good job in improving performance. We have included this in the paper in Section 4 now.
>
> We have now updated the discussion on the distinction between ergodic MDPs and ergodic rewards and added the following paragraph:
>
> “Generally, an ergodic MDP does result in a stationary state distribution. The reward function is, typically, a deterministic function from the state (and potentially action) space to the real numbers. Thus, also the reward distribution is stationary. While an ergodic stochastic process is stationary, the converse does not necessarily hold. The conditions under which stationary processes are ergodic have, for instance, been discussed by Parzen (1958).”

---

### Review · Reviewer_2U9z · 2024-11-11

**Summary Of Contributions:**

This paper investigated the context where optimizing expectation of returns could result in unsafe policies attaining high return in rare cases but fail most of the time. One of the potential causes could be non-ergodic rewards and authors investigated it using the coin toss example. Through eqs 4 and 5 the authors explained the ergodic rewards maximizing the expectation of which is equivalent to maximizing growth rate.

**Audience:**

Yes

**Claims And Evidence:**

No

**Requested Changes:**

Please refer to the above my requested changes.

**Strengths And Weaknesses:**

**Strengths**:\
The paper is in general well-written with a clear motivation. The authors nicely explained the issue of non-ergodic rewards with plain words and a toy example, backed with sound theory in the appendix. I believe the paper is of interest to the audience in safe/robust RL and ergodic RL and I learned from it.

**Weakness**:\
One apparent weakness is the lack of convincing experimental results. As the proposed solution is simple (logarithmic/learned variance), it should not be difficult to test them on more sophisticated algorithms and environments that come with the stable-baselines library. If compute resources are of concern, then I believe the authors should compare with the baseline case in coin toss with $\tilde{r}(t_k) = R(t_k) - R(t_{k-1})$. This baseline gives the agent a sense of reward difference and we should see it improves learning but still underperforms the proposed log-difference if the theory is correct.

Moreover, it is quite unclear to me how Cart-pole and Reacher in Figure 4 relate to the central theme of the paper, as their rewards do not have "the loss" part as the coin toss example does. The authors did not explain why the original REINFORCE fail on reacher either. In my opinion, to strengthen this part, it may be wiser to try a modified reward function that explicitly reflects such non-robustness; or just remove Figure 4 totally and replace with theoretical insights/discussion from the appendix.

---

> ### Author Response · Authors · 2024-11-25
>
> We thank the reviewer for the positive evaluation of our work. We first would like to comment that we are actually training PPO using $r(t_k) = R(t_k) - R(t_{k-1})$ in the coin-toss example. We have made this more clear now in the text.
>
> In this paper, we combine our developed transformation with a Monte Carlo-based RL algorithm (REINFORCE). We stress this now more in the revised version. In particular, we provide pseudocode showing how our algorithm is trained. The algorithm always uses a collected time-series of returns to learn the transformation. For the combination with other RL algorithms from the stable baselines, the transformation would instead need to be computed in an incremental fashion. Such an application is certainly a highly relevant topic for future research, as we also mention in the conclusions, but beyond the scope of this paper.
>
> A main goal of our algorithms is to increase robustness against heavy-tailed reward distributions. As we motivate in Section 7, with this we also gain robustness to model uncertainties. Thus, in Section 7, we show that our algorithm improves upon the baseline in settings where the environment dynamics change between training and testing, thus, showing greater robustness. Besides, the reward structure of the cart-pole system is also non-ergodic in that there exists an absorbing state: the episode ends if the pole drops. We motivate both aspects now more clearly in the revised version of the paper.

---

### Review · Reviewer_cFTQ · 2024-11-18

**Summary Of Contributions:**

The paper considers a reinforcement learning setup with non-ergodic rewards. This setup is closely related to Risk Aware RL where algorithms and their analysis are tuned to guarantee performance of any RL agent instead of average performance of large number of RL agents. This setup is particularly important with heavy tailed reward distribution where optimizing over average cases may be detrimental of the larger number of agents. It can find use cases in RL based LLM alignments to ensure average behavior from head tokens does not dictate behavior of long tail tokens.

**Audience:**

Yes

**Claims And Evidence:**

Yes

**Requested Changes:**

In addition to the weaknesses above:

1. In the paragraph below Eq. 8: "Logarithmic returns (or regrets) are common in the RL literature.". Please cite sources for this.

2. It would help to lay out a proper step by step algorithm for "Ergodic"-RL. You can use a meta algorithm which estimates $\mu^2(u)$, transforms the rewards, and then applies off the shelf RL algorithms (PPO/REINFORCE), etc.

3. After multiple re-reads, I finally understood the objective is to construct a transformation such that the increments grow linearly and hence simply taking $R'(t+1) = R(t+1) - R(t)$ does not work. Please specify this discussion appropriately in the abstract and introduction to strengthen the paper and also simulate the toy examples with this to further strengthen the result.

**Strengths And Weaknesses:**

Strengths:

1. The considered problem of heavy tailed rewards distribution from the lens ergodicity is interesting. It is good to motivate such a discussion in the community.

2. The paper presents a variance stabilizing transform as a mechanism to fix the problem of non-ergodic rewards. Further, the paper shows with simulations that the proposed approach works.


Weaknesses:
1. I could not find why the setup is non-ergodic (in rewards). To convince myself, I verified the result with simulations, I realized that the multiplicative scaling of rewards R(t) = f(R(t-1)) was leading to non-ergodicity here. Especially, after taking the logs of the rewards, it was much clearer that the expected value of the cumulative sum log of the reward function is negative, but the cumulative sum function is positive. Adding some details will help readers better understand the simulation and the problems with non-ergodic rewards.

2. How is equation 6 derived for the discrete time case? The paper first need to provide the transformation setup of Peters & Adamou (2018) to motivate the Equation 6.

3. In an RL setup, R(t_k) is dependent on the policy. However, no such policy is mentioned in the expectation computation of $\mu^2(u)$. How/Why does the proposed transformation work for any mis-specified policy based expectation? This is the biggest challenge I think this work faces. See example of variance measured for a particular policy here:

La, Prashanth, and Mohammad Ghavamzadeh. "Actor-critic algorithms for risk-sensitive MDPs." Advances in neural information processing systems 26 (2013).

4. What are the challenges in studying the gap between the expected untransformed rewards and expected transformed rewards is not clear? Also is their a gap on using transformation for any arbitrary policy?

---

> ### Author Response · Authors · 2024-11-25
>
> We thank the reviewer for the positive evaluation of our work and the constructive feedback.
> To make the non-ergodicity of the coin-toss game more intuitive, we have now added a new figure (Figure 1b) that illustrates all possible sample paths for three time steps. There, it can be seen that while the average across paths grows, there is only one out of 4 possible paths that leads to a higher than the initial return. In that way, as we let time go to infinity, the average across paths will grow while the set of paths where the final return is larger than the initial one will shrink to a set of measure 0.
>
> We have now adapted the explanation around Equation (6) and, in particular, explained why the increments of the transformed returns should follow a Brownian motion.
>
> In the coin-toss game, we have a setting where different policies amount to different choices of $F$, which does not change the problem structure. Thus, learning a single transformation is sufficient. However, in the more general RL case, this may not be true. We have accounted for this case now more explicitly in the revised version of the paper. In particular, in Section 4, we now make explicit that we show an algorithm for estimating a transformation for a single time-series of returns, i.e., based on one particular policy. Then, in Section 7, we explain how we embed this into an RL algorithm such as REINFORCE. For this, we have now also added pseudocode in Algorithm 1, as suggested by the reviewer. We are now also discussing the suggested paper.
>
> We are happy to expand on this in case it does not answer all comments of the reviewer. In particular, we do not claim that the transformation will work for any potential policy. There may be policies or environments where finding an ergodicity transformation is not possible. In such cases, the proposed transformation will try to get as close as possible to an ergodicity transformation.
>
> We have added references for the statement about logarithmic returns and regrets. Further, we added a statement about the linear behavior of transformed returns in the introduction.

---

### Decision · Action_Editor_Jcmp · 2024-12-19

**Recommendation:** Accept with minor revision

**Comment:**

This paper is a resubmission of TMLR 2783. The manuscript has been improved overall, especially including a more comprehensive discussion on how the work relates to previous work in risk-sensitive RL and how the concept of reward ergodicity relates to the most common MDP ergodicity.

The reviewers, beside noting some area for improvements over the clarity of the manuscript, generally agree that this paper is a valuable contribution to TMLR and advocate for accepting it.

My recommendation is to accept the paper upon a few additional changes to be integrated in the final version:
- The title could be modified to give a clearer account of the paper's contribution. This is ultimately the authors choice, but something on the lines of "Reinforcement learning with non-ergodic rewards: Robustness via ergodicity transformation" could be considered;
- The contribution of the paper is in general very tied to the coin-toss problem. The problem per se is interesting, but the paper could clarify early on (e.g., in the abstract) that this is the main object of the analysis. Moreover, it is not as established as, e.g., LQR, so a description of which applications may benefit from solving this problem would be useful;
- Others:
    * Figure 1 only shows "bad trajectories", which does not "visually" validate the reported expected return;
    * Risk-sensitive RL section looks underdeveloped. For the exponential utility transformation, it does not clarify which $\beta$ values have been tested. Is there a value of $\beta$ for which $F \in (0, 1)$? Other approaches in the risk-sensitive RL literature seems to potentially solve the coin-toss problem, such as the mean-variance mentioned in the paper and CVaR optimization (e.g., https://proceedings.neurips.cc/paper/2015/hash/024d7f84fff11dd7e8d9c510137a2381-Abstract.html)

**Audience:**

The paper tackles a setting that is relevant to the risk-sensitive RL community and possibly interesting for a broader RL community.

**Claims And Evidence:**

The paper tends to present broad claims that are mostly supported in one single (yet very relevant) case, i.e., the coin-toss example. This is not a major issue, but some claims could be toned down.

---

> ### Author Response · Authors · 2025-01-10
>
> We thank the AE for handling our paper and the positive evaluation of our work. We have uploaded the camera-ready version, where we have addressed the remaining comments:
>
> We have changed the title to "Reinforcement learning with non-ergodic reward increments: robustness via ergodicity transformations" to reflect our contributions better; we mention the "instructive example with heavy-tailed returns," i.e., the coin-toss example, in the abstract and discuss its relevance in section 6.2; we have included in the appendix more simulation results for the coin-toss examples that show some good trajectories as well to validate the expected value visually; we discuss more explicitly the selection of $\beta$; and we included the paper the reviewer mentioned in our related work section 6.3.